# Oxford brain health clinic: protocol and research database

Melissa Clare O'Donoghue [1,2] Jasmine Blane,[1,2] Grace Gillis [1,2] Robert Mitchell,[2] Karen Lindsay,[1,2] Juliet Semple,[1,2] Pieter M Pretorius,[3] Ludovica Griffanti,[1,2] Jane Fossey,[2,4] Vanessa Raymont [1,2] Lola Martos,[1,2] Clare E Mackay [1,2]

¹Department of Psychiatry, University of Oxford, Oxford, UK
²Oxford Health NHS Foundation Trust, Oxford, UK
³Oxford University Hospitals NHS Foundation Trust, Oxford, UK
⁴College of Medicine and Health, University of Exeter, Exeter, UK

**Correspondence to**
Professor Clare E Mackay;
clare.mackay@psych.ox.ac.uk

## ABSTRACT

**Introduction** Despite major advances in the field of neuroscience over the last three decades, the quality of assessments available to patients with memory problems in later life has barely changed. At the same time, a large proportion of dementia biomarker research is conducted in selected research samples that often poorly reflect the demographics of the population of patients who present to memory clinics. The Oxford Brain Health Clinic (BHC) is a newly developed clinical assessment service with embedded research in which all patients are offered high-quality clinical and research assessments, including MRI, as standard.

**Methods and analysis** Here we describe the BHC protocol, including aligning our MRI scans with those collected in the UK Biobank. We evaluate rates of research consent for the first 108 patients (data collection ongoing) and the ability of typical psychiatry-led NHS memory-clinic patients to tolerate both clinical and research assessments.

**Ethics and dissemination** Our ethics and consenting process enables patients to choose the level of research participation that suits them. This generates high rates of consent, enabling us to populate a research database with high-quality data that will be disseminated through a national platform (the Dementias Platform UK data portal).

## STRENGTHS AND LIMITATIONS OF THIS STUDY

⇒ The Oxford Brain Health Clinic (BHC) embeds high-quality assessments into routine clinical care for typical patients with memory problems.
⇒ The BHC MRI protocol is aligned with the UK Biobank providing a unique opportunity to link the power of big data and individual patients at the clinical interface.
⇒ The BHC ethics and consenting process, designed in partnership with an active patient and public involvement advisory group, enables patients to choose the level of research participation that suits them.
⇒ The BHC Research Database and associated information governance will facilitate research use of real-world clinical data sharing where consent is given.
⇒ Some elements of the BHC model, particularly the MRI, are hard to scale up without substantial changes in commissioning for memory clinics.

## INTRODUCTION

In the UK, adults over the age of 65 years who visit their general practitioner (GP) with concerns about memory are typically referred to psychiatry-led memory clinic services. In 2009, the Memory Services National Accreditation Programme was set up by the Royal College of Psychiatrists to create a quality improvement and accreditation network for services that assess, diagnose and treat dementia in the UK. Despite this, the assessments available to memory clinical services to inform diagnosis in have remained largely unchanged for decades. Clinical services also continue to focus largely on the diagnosis of established dementia, despite growing understanding of risk factors, biomarkers and management of early neurodegenerative diseases that can lead to dementia. Novel

therapeutics for early disease are now potentially imminent,[1–3] and there is good evidence that personalised risk reduction can improve outcomes.[4 5] Services urgently need updating to be able to adequately stratify patients and deliver such interventions.[6 7]

Meanwhile, research into those same risk factors, biomarkers and novel interventions usually takes place in academic settings, where studies are typically conducted in research cohorts recruited by accessing clinical populations. This set-up is more common in neurology-based clinical settings where the patients are on average younger and have different symptoms profiles to those seen in psychiatry settings. For example, one of the largest cohorts derived by embedding research in a clinical setting is the Amsterdam Dementia Cohort, with an average age of 65;[8] and the most commonly cited dementia cohorts used in biomarker (particularly imaging) research are the Alzheimer's Dementia Neuroimaging Initiative (ADNI)

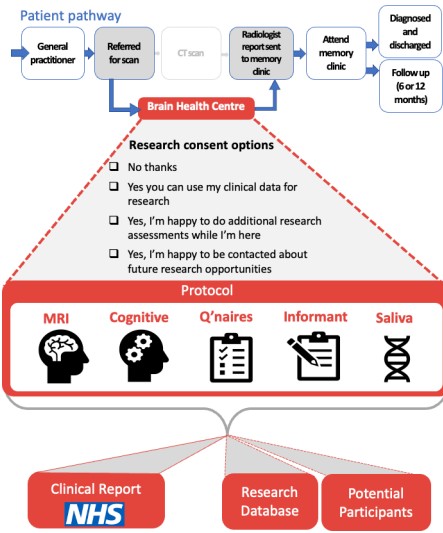

**Figure 1** Overview of the Brain Health Clinic patient pathway and data flow.

series of cohorts (ADNI 1, 2, 3) with average ages in the low 70s.[9–11] In contrast, a 2019 audit of UK memory services showed the average age of attending patients was 79 years, including referred patients under the age of 65.[12]

There is thus a gap to provide both improved diagnosis and prognosis for all patients presenting to memory clinics, and to enable data gathering and research in a real-world cohort that is representative of the population of patients who are presenting to memory clinics.

### Oxford Brain Health Clinic pilot

The Oxford Brain Health Clinic (BHC, see figure 1) is an ambitious and innovative joint clinical-research service that aims to prepare memory clinics for the future of dementia diagnosis and treatment at the same time as creating a platform for development and evaluation of novel diagnostics and therapeutics. Funded by the National Institute of Health Research (NIHR) Oxford Health (OH) Biomedical Research Centre (BRC) and the NIHR Cognitive Health Clinical Research Facility, the BHC enhances assessments available to patients with memory problems by providing access to high-quality assessments not routinely available in clinical practice (eg, MRI rather than CT brain scans). Enhanced information is fed into clinical notes, improving the quality of information available to clinicians when making diagnoses in the memory clinic.

All patients and their accompanying relative attending the BHC are invited to participate in research, either by consenting to the use of clinical data for research, by completing additional research assessments and/or choosing to be contacted about future research opportunities. We hypothesise that this integrated and equable access to research participation will enable us to exceed the target of the Prime Minister's Challenge on Dementia to have 10% of dementia patients involved in research.[13] By embedding research in the clinical service, the BHC

provides a translational interface to develop and evaluate new approaches to diagnosis, risk reduction, treatment and prevention in real-world patients and in turn enable new advances to be rapidly implemented in clinical practice to improve patient care.

The BHC pilot was launched in August 2020, aiming to demonstrate feasibility, practicability, scalability and the benefits that the clinic can offer long term. Here we describe the BHC Research Database, a repository of real-world data and trial-ready volunteers, and the data collection protocol. We present preliminary data from the first 16 months of referrals and report the rate of research consent.

## METHODS AND ANALYSIS
### Design

The BHC Research Database stores and makes available data collected at the OH BRC BHC. Patients with memory problems referred by their GP to pilot-partner memory clinics in OH NHS Foundation Trust (FT) can be referred to the BHC for assessments prior to their memory clinic appointment. Instead of receiving a standard CT brain scan, patients attending the BHC receive an MRI scan, as well as cognitive assessment and questionnaires as part of their clinical assessments. To support MRI safety screening, patients are accompanied to their appointment by a relative or friend, who also completes an informant interview as part of the clinical assessment. At the BHC appointment, patients and their accompanying relatives/friends are invited to join the BHC Research Database (described further below), and complete additional research assessments, including further MRI scanning and saliva sample.

BHC appointments currently last up to 2.5 hours, including all NHS and research assessments. The clinical portion takes around 1.75 hours to complete. At the end of the clinic, staff summarise clinical information in a BHC clinical report, which is uploaded into the Trust's electronic patient records system and used in the subsequent memory clinic appointment to aid clinical decision-making and diagnosis. Clinical MRI scans are reported by a neuroradiologist. Other research information can be shared in the clinical report if requested by the patients' memory clinic doctor.

The Oxford BHC takes place in the Oxford Centre for Human Brain Activity, a University of Oxford (OU) site and part of the Wellcome Centre for Integrative Neuroimaging. The BHC operates as part of the NIHR cognitive health Clinical Research Facility. All BHC staff are either OH employees or University employees with honorary contracts from OH.

### Patient and public involvement

People with lived experience of dementia have been integral partners in establishing both the protocol and research database for the BHC. Our advisory panel includes people living with dementia, carers and

interested members of the public. Together with our steering group lay member, they have provided vital ideas and feedback, as well as connections to wider networks, such as carer groups, that offer further lived experience to enhance the BHC. For example, lay contributions have transformed the format and language of research information for patients and carers, and participated in 'trial runs' of the clinic to provide feedback on patient journey. The BHC advisory group also codeveloped a set of strategic objectives for public involvement with the BHC, establishing the infrastructure to embed patient and public involvement (PPI) in the BHC and facilitate continued collaboration with our public partners. This is to ensure the BHC is directed by the needs and concerns of the people affected by memory problems and dementia.

## Participant selection

Partner memory clinics in OH NHS FT receive referrals from primary care, which are reviewed by a duty psychiatrist. This usually involves a phone call as well as a review of notes. There are no formal inclusion/exclusion criteria. All patients requiring imaging are referred to the BHC for an MRI scan unless the duty psychiatrist has reason to believe they would not be able to undergo an MRI scan. Reasons include clear contraindications to MRI (implanted devices, metallic foreign bodies, eye injuries, or exceeding the size and weight limitations) and/or patients having mobility problems (limited ability to self-transfer onto the scanner or to lie flat) or being too physically frail to tolerate the length of the BHC appointment. Patients who require imaging but cannot be referred to the BHC are offered a CT scan as standard.

Patients referred to the BHC complete MRI safety screening via telephone with a radiographer. Individuals with no MRI contraindications are scheduled for an appointment, and are sent a summary research information sheet along with their appointment letter, including the following:

► A brief description of the BHC Research Database.
► A brief description of what joining the BHC Research Database would involve.
► Contact information to discuss the BHC Research Database further.
► A clear statement that they are not obliged to join the BHC Research Database, and declining to join the BHC Research Database will not affect their clinical care.

Patients also receive a reminder call the day before the BHC appointment.

## Consent

At their BHC appointment, patients are provided with a full information sheet and have the opportunity to ask any questions. Informed written consent is obtained prior to any research procedures being undertaken.

Patients attending the BHC can consent to take part in research in three ways:

► Consent for use of clinical data for research: patients agree for clinical data collected at the BHC for their NHS assessment, along with relevant information from their medical notes (eg, diagnosis and medication), to be stored and made available in the BHC Research Database (described further below).
► Consent to additional research assessments: at their BHC appointment, patients can complete additional assessments for research purposes only (described below). The results of these assessments are stored and made available in the BHC Research Database. Patients are able to select which, if any, additional research assessments they wish to complete.
► Consent for research recontact: patients can agree for their contact information to be stored for the purpose of recontacting them about future research opportunities. Patients may be contacted on the basis of characteristics stored in the BHC Research Database (eg, cognitive score, hippocampal volume) to provide run-in data for clinical trials. Patients may request that the person accompanying them to their BHC appointment is contacted on their behalf about future research opportunities.

Patients who consent to additional research assessments or consent for research recontact are required to consent to the use of clinical data for research. As the clinical assessments form part of patients' routine NHS care, these are conducted whether or not the patient chooses to join the Research Database.

Patients' capacity to consent to take part in research is assessed prior to consent being taken. If a patient lacks capacity to consent to research, their accompanying relative or friend is able to act as a consultee to agree to research participation on the patient's behalf.

The person accompanying the patient to the BHC is also offered the opportunity to take part in research themselves, by completing research questionnaires and giving consent for research recontact.

## Procedures and outcome measures

All patients attending the BHC complete NHS assessments, consisting of cognitive assessment, questionnaires and clinical MRI brain scan. The accompanying relative completes an informant interview providing corroborative information about changes in the patient's memory, mood, daily life and social circumstances. For patients that consent to use of clinical data for research, the results of clinical assessments are stored and made available via the BHC Research Database, as well as being communicated to the memory clinic via the BHC clinical report.

Additional research assessments that patients and relatives can consent to currently include research MRI sequences (*patient*), saliva sample (*patient*) and additional questionnaires (*accompanying relative*). The results of research assessments are also stored and made available via the BHC Research Database.

## Cognitive assessment

Patients complete the Addenbrooke's Cognitive Examination[14] (ACE-III) which assesses five cognitive domains: attention, memory, language, verbal fluency and visuospatial function. Assessments take approximately 25–30 min.

## MRI

Patients are scanned on the 3T Siemens Prisma scanner at the Oxford Centre for Human Brain Activity, Oxford, using a 32-channel head coil. Patients must be accompanied to their BHC appointment by someone with knowledge of their medical history to support MRI safety screening prior to the scan. Patients' height and weight are also measured prior to the scan.

The clinical MRI scan protocol (~15 min) includes a 3D diffusion-weighted image, fluid-attenuated inversion recovery image, high-resolution T1 structural image and susceptibility-weighted images. Clinical images are pushed to clinical imaging systems (Picture Archiving and Communication Systems) and are reported by a neuroradiologist using a standardised framework for qualitative reporting developed jointly with the BHC. This framework includes important negatives (eg, tumour, hydrocephalus), atrophy, white matter hyperintensities, microhemorrhages, infarcts/intracerebral haemorrhage and other clinically relevant incidental findings.

The research MRI protocol (~20 min) includes pseudo-continuous arterial-spin labelling, multi-shell diffusion-weighted imaging and resting-state functional MRI. Patients that consent to complete research sequences remain in the scanner after their clinical scans once the radiographer has confirmed they are still happy to continue.

Where possible, sequences (both clinical and research) have been matched to those used in the UK Biobank[15] (UKB) to facilitate future use of the eventual 100 000 UKB brain scans as normative data against which to compare data from BHC patients. Images from patients that consent to use of clinical data for research and/or complete the research MRI are processed by a modified version of the UKB image processing pipeline,[16] producing the same set of imaging-derived phenotypes as are available from the UKB.

## Questionnaires

Patients are sent a set of paper questionnaires with their appointment letter and are asked to complete these prior to the appointment and bring them to the BHC. Questionnaires include measures of depression (Patient Health Questionnaire-9[17]), sleep (Pittsburgh Sleep Quality Index[18]), physical activity (Short Active Lives Questionnaire[19]), alcohol use (Single Alcohol Use Screening Questionnaire[20]) and long-term health conditions (Long-Term Conditions Questionnaire—short form[21] (LTCQ-8)).

Accompanying relatives that consent to complete research questionnaires can complete both subjective well-being measures (Relative Stress Scale[22]) and informant-based measures of patient cognitive change (Informant Questionnaire on Cognitive Decline in the Elderly[23]) and neuropsychiatric symptoms (Neuropsychiatric Inventory[24]). The patient must also provide consent for the accompanying person to complete informant-based measures.

## Saliva

Samples are collected using an Oragene DNA Self Collection Kit (DNA Genotek, Ontario, Canada). DNA will be extracted and used for apolipoprotein E (APOE) genotyping and whole-genome sampling.

## Clinical observations

Staff complete the Rockwood Clinical Frailty Scale,[25] a global clinical measure of frailty evaluated by a clinician and rates patients fitness/frailty on a 9-point Likert scale (1=least frail, 9=most frail). Staff also write a brief summary of any clinical observation during the appointment, such as behaviour, appearance, mood and insight. The clinical observation summary is currently only used clinically, and not for research.

## Informant interview

The person accompanying the patient completes an informant interview with a member of staff, including questions about changes in the patient's cognition, mood, behaviour and function, and current social circumstances. The qualitative informant report, uploaded to electronic patient records for use in the memory clinic appointment, is currently only used clinically and not for research purposes.

## Sample size

As a pilot using a convenience sampling approach, and creating a research database rather than conducting a research study addressing a particular hypothesis, it was not possible to conduct a sample size calculation. Recruitment is ongoing.

The BHC received 157 referrals from the launch in August 2020 until November 2021. Of these, 108 attended the BHC, 15 were scheduled for future appointments, and 34 referrals were returned to routine NHS memory services prior to attendance. Of these returned referrals, 12 were due to MRI incompatibility (eg, claustrophobia (n=4), metalworks (n=1), possible MRI screening inaccuracy (n=5), inability to lie flat (n=1), weight (n=1)), 10 refused MRI scan, and 12 were referred back for other reasons (eg, mobility and transportation issues (n=3), hospitalisation (n=2), inappropriate referral (n=1), appointment no longer required (n=4), unable to contact (n=2)).

MRI scans were well tolerated by BHC patients. Of 108 attendees, 103 (95.4%) were able to be scanned (two not scanned due to inability to lie in scanner, two had safety contraindications on the day, one was claustrophobic). One hundred patients (92.6%) completed the full clinical imaging protocol (three scans were abandoned due to claustrophobia and discomfort in the scanner).

**Table 1** Uptake of research by patients

| | Use of clinical data for research | Additional research assessments | | | | Recontact about future research |
| --- | --- | --- | --- | --- | --- | --- |
| | | Any additional assessment | MRI | Saliva | Informant questionnaire | |
| N (%) | 101 (93.5%) | 93 (86.1%) | 69 (63.9%) | 77 (71.3%) | 88 (81.5%) | 79 (73.1%) |

% of total patient attendees, n=108.

Uptake of research at the BHC (summarised in table 1) has been high, as shown in figure 2. Of the 108 attendees, 94% (n=101) consented to use of clinical data in the BHC Research Database. These patients were on average 78.3 years old (65–101), 50.5% were female, and had average ACE-III scores of 72.9 out of a maximum 100 (9–98). As shown in figure 3, the majority of patients were in their mid-70s and mid-80s. ACE-III scores were variable with 81% (n=79) scoring 88 or fewer and 67% (n=66) scoring 82 or fewer.[14] Full demographics are shown in table 2.

Eighty-six percentage (n=93) of attending patients also consented to complete additional research assessments at their appointment. Sixty-four per cent (n=69) agreed to additional research MRI, 71% (n=77) consented to provide a saliva sample and 81% (n=88) consented to their relative completing informant questionnaires. Consent and completion rates of additional patient assessments are shown in figure 4. Only 105 relatives had the opportunity to consent to research participation (one patient attended alone and two relatives were not interviewed due to staff shortages). Eighty per cent (n=84) of accompanying relatives consented to complete additional research assessments, and 77% (n=81) of accompanying relatives completed the informant questionnaires.

Seventy-three per cent (n=79) of attending patients consented to be recontacted about future research as did 72% (n=76) of accompanying relatives. 15.7% (n=17) of patients requested a relative be contacted on their behalf about future research opportunities.

### Data analysis plan

As a research database rather than a research study, the BHC Research Database is not designed around a specific research question or hypothesis. Instead, the BHC and the research database provide a platform for multiple research studies and trials.

Examples of the research that is already underway includes (1) description of the MRI and cognitive characteristics of a representative memory clinic population, (2) the clinical translation of UKB image analysis pipelines, (3) application of novel cognitive and digital biomarkers, (4) development of radiological decision support tools, (5) health economics evaluation of the BHC model and (6) evaluation of patient and clinician experience, including qualitative research. The database is also being used to approach patients about participating in PPI activities to support future research plans.

### ETHICS

The BHC Research Database was reviewed and approved by the South Central—Oxford C research ethics committee (SC/19/0404).

### DISSEMINATION

By making real-world data and trial-ready volunteers available to the scientific community, the BHC Research Database aims to facilitate and actively encourage collaborative and transparent research. Shared data can be used in research to increase understanding of diseases that lead to dementia, as well as to improve diagnostics, prognostics, prevention and treatments available for dementia.

Our very high research consent rates give rise to a highly inclusive and representative cohort, and by aligning our imaging and genetic analysis with the UKB we can make direct comparisons to the largest population database in the world.

### Data management

All BHC data is managed in a bespoke clinical database, created using Exprodo software (www.exprodo.com). The BHC clinical database, used to schedule appointments and record data collected during appointments, sits within the OH NHS network behind a firewall.

Based on the consents provided, contact details and deidentified research information are pushed to the BHC Research Database. The research database consists of three separate databases:
► Research DB: containing data from the clinical database and relevant information from medical notes to be used in research with all identifying information removed (deidentified).

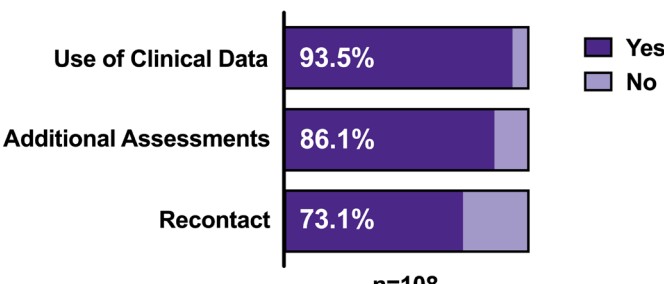

**Figure 2** Patient consent rates for each of the three research options offered at the Brain Health Clinic.

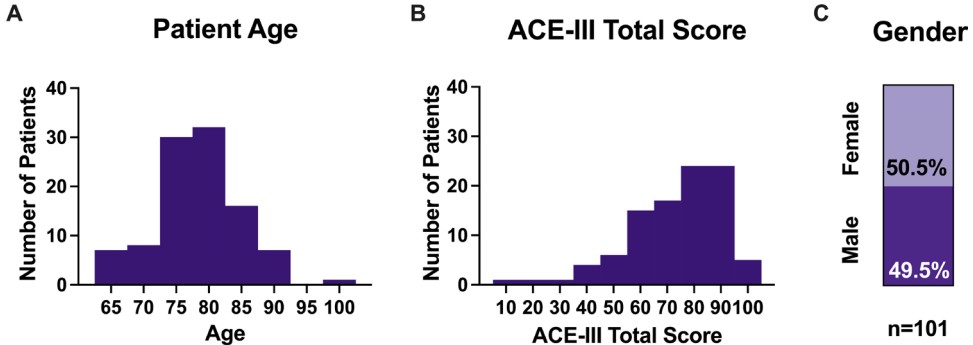

**Figure 3** Demographics of patients consenting to join the Brain Health Clinic Research Database. (A) Patient age distribution. (B) ACE-III total score distribution. (C) Proportion of males and females. All figures shown for patients consenting to use of clinical data for research (n=101). ACE-III, Addenbrooke's Cognitive Examination-III.

► Recontact DB: containing names, address, email, contact preferences for patients and volunteers that consented to be recontacted about research (or to receive a newsletter).
► Imaging DB: specialist imaging database holding imaging data (eg, DICOM, NIFTI) for clinical (where patient consented to use of data for research) and research MRI scans.

The Research and Recontact DBs also use Exprodo software while the Imaging DB uses XNAT software. All three research databases are held on University servers.

### Data governance

Information security and governance is managed by the information governance teams in the University and NHS Trust and governed by data privacy impact assessments and third-party security assessments. The BHC Research Database is also governed by OU, OH and BHC-specific data governance, security, management and access policies. All staff handling BHC Research Database data are trained in the principles of Information Governance, the Data Protection Act and the EU General Data Protection Regulation.

### Data sharing

The BHC will provide access to research data to bona fide researchers for health-related research that is in the public interest. Requests for sharing of de-identified data and/or access to BHC patients consented for recontact will be considered by the BHC data access committee, which includes PPI contributors.

### Research data

De-identified imaging and non-imaging data stored in the Research DB and Imaging DB will be available through the MRC Dementias Platform UK (DPUK) infrastructure (https://www.dementiasplatform.uk). Data will be accessible via the DPUK data portal.[26] DPUK data access policies and procedures will apply to access the BHC research data. Researchers wishing to gain access to data from the BHC Research Database must agree to the terms and conditions of the access, including acknowledgement of the BHC Research Database and the OH BRC.

### Recontact

The BHC Research Database includes a registry of patients and their relative/friends who have consented to be recontacted about future research studies. Researchers can apply to the BHC Research Database for potential participants for their studies. Researchers who wish to access BHC participants must complete an online project application form, with approval based on evidence of ethical approval, funding and the project falling in the remit of the BHC (dementia or brain health research).

| Table 2 | Summary of demographics | |
|---|---|---|
| Age, mean (range) | 78.3 (65–101) | 66.3 (37–87) |
| Female, N (%) | 51 (50.5%) | 45 (61.6%) |
| Age at leaving full-time education, mean (range) | 18.5 (12–42) | – |
| ACE-III total score, mean (range) | 72.9 (9–98) | – |
| Rockwood Frailty Score, mean (range) | 2.68 (1–7) | – |
| Lacked capacity, N (%) | 16 (15.8%) | – |

Patient figures reported for those consenting for use of clinical data for research (n=101). Missing data: three missing age leaving full-time education and one missing Rockwood Frailty Score; three missing ACE-III score. Relatives figures reported for those who consented to be recontacted about future research and completed questionnaires (n=73).
ACE-III, Addenbrooke's Cognitive Examination-III.

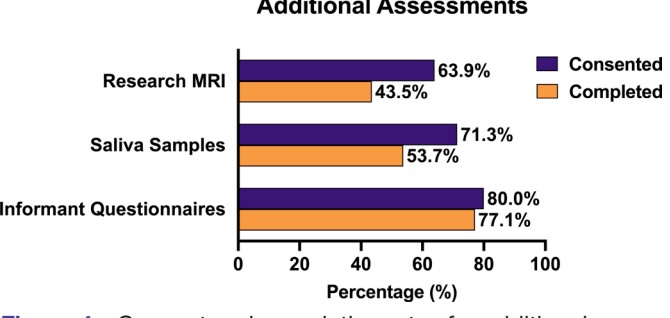

**Figure 4** Consent and completion rates for additional patient research assessments.

Researchers can choose to recontact volunteers based on variables included in the Research DB, such as APOE genotype, hippocampal volume or ACE-III score. If a search for volunteers returns a total sample of less than 10 participants, the researcher will not be able to proceed with the search, in order to protect the identity of patients within the research database. Volunteers meeting the specified criteria will be informed of the research opportunity by the BHC and then, if they are interested in taking part, follow-up directly with the invited study researchers. Recruiting researchers do not have access to any volunteer contact information until the volunteer chooses to hear more about the research opportunity. Volunteers will be required to complete a study specific consent form for any research they choose to participate in.

All researcher and project applications and volunteer searches will be reviewed and, if appropriate, approved by the database administrator and monitored by the data access committee. Researchers must agree to not store or use contact information for any purpose other than the approved study, and not to share contact information with any third party.

**Acknowledgements** We are grateful to the operations team of the BHC including: Amanda Colston, Clare Hamblin, Emily Johnson, Rebecca Williams, Nicky Watkins, Jessica Wallis, Nicola Aikin, Jon Campbell, Sophie Walker, Leona Wolters, Michael Ben-Yehuda, Emma Craig, Gary Gibbs, Karla Westphal, Rebecca Smith, Deborah Wilkinson, Gemma Butler, Aurelija Burbaite, Julia Hamer Hunt, Qi Pei, Becci Dow, Candy Stone, Sebastian Rieger, Paul Semple and Shona Forster. The views expressed are those of the author(s) and not necessarily those of the NIHR or Department of Health and Social Care.

**Contributors** CEM led the development of the BHC, designed and drafted the protocol and manuscript. MCO project-managed the development of the BHC, designed and drafted the protocol and manuscript. GG conducted data analysis and drafted the manuscript. JB, RM, KL, JS, PMP, LG, JF, VR and LM all contributed to the design of the protocol and reviewed the protocol and manuscript.

**Funding** This work was supported by the NIHR Oxford Health Biomedical Research Centre (grant number n/a), a partnership between the University of Oxford and Oxford Health NHS Foundation Trust, the NIHR Oxford Cognitive Health Clinical Research Facility and the Wellcome Centre for Integrative Neuroimaging. The Wellcome Centre for Integrative Neuroimaging is supported by core funding from the Wellcome Trust (203139/Z/16/Z). For the purpose of open access, the authors have applied a CC-BY public copyright licence to any Author Accepted Manuscript version arising from this submission. LG is supported by an Alzheimer's Association Grant (AARF-21-846366).

**Competing interests** CEM is a cofounder and shareholder of Exprodo Software, which was used to develop the BHC database. CEM serves on a Biogen Brain Health Consortium (unpaid). No other competing interests to report.

**Patient and public involvement** Patients and/or the public were involved in the design, or conduct, or reporting, or dissemination plans of this research. Refer to the Methods section for further details.

**Patient consent for publication** Not applicable.

**Provenance and peer review** Not commissioned; externally peer reviewed.

**ORCID iDs**
Melissa Clare O'Donoghue http://orcid.org/0000-0003-0819-2513
Grace Gillis http://orcid.org/0000-0002-6123-3647
Vanessa Raymont http://orcid.org/0000-0001-8238-4279
Clare E Mackay http://orcid.org/0000-0001-6111-8318

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
