## [Reviewer comments · BMJ Open]

ARTICLE DETAILS

TITLE (PROVISIONAL)	The Oxford Brain Health Clinic: Protocol and Research Database
AUTHORS	O'Donoghue, Melissa; Blane, Jasmine; Gillis, Grace; Mitchell, Robert; Lindsay, Karen; Semple, Juliet; Pretorius, Pieter M; Griffanti, Ludovica; Fossey, Jane; Raymont, Vanessa; Martos, Lola; Mackay, Clare

VERSION 1 – REVIEW

REVIEWER	Pemberton, Hugh G. University College London, Queen Square Institute of Neurology
REVIEW RETURNED	02-Dec-2022

GENERAL COMMENTS	This is a well written, timely and relevant study protocol paper. The authors have highlighted the limitations in current assessments for patients with memory concerns and clearly documented their plan to address them. Of particular relevance is the drive to provide MRIs rather than CT scans, where possible. The manuscript could be published in its current state but I have a few comments where further detail or clarity may be beneficial to readers, assuming the authors are willing to make these few changes. Introduction References 1 and 2 could now also include the recent NEJM lecanemab paper (https://www.nejm.org/doi/pdf/10.1056/NEJMoa2212948). Methods and Analysis Carer groups are mentioned, are there specific plans to run a support group(s) as part of the Oxford BHC? “Patients who require imaging but cannot be referred to the BHC are offered a CT scan as standard” – What sort of patients cannot be referred to the BHC? In this context, is this just those contraindicated, too frail, or unable to travel for MRI? Please clarify. I note the modified version of the UKB image processing pipeline, could the authors provide more detail here, i.e. will the MRIs be automatically post processed by any research (or commercial) software for further analysis? i.e. WMH quantification, volumetry etc. Will these results be added to the research database? Very encouraging MRI tolerance levels and research uptake for the pilot patients! Do the authors have any comments as per the high research uptake levels and what can be done to replicate this in other initiatives?
---

	The authors state in the “data sharing” section, that all data will be made available pending request approval etc. It could be useful to state this earlier in the paper (perhaps expand on “article summary” bullet point 4? This currently does make clear that the database will be made available etc.) and/or reference this section earlier, as I suspect this will be of great interest to most readers.
--	--

REVIEWER	Mukaetova-Ladinska, Elizabeta University of Leicester, neuroscience, Psychology and Behaviour
REVIEW RETURNED	21-Feb-2023

GENERAL COMMENTS	Well and extensively presented protocol for engaging dementia sufferers in research, creation of data base, clinical and neuroradiological investigations, genotyping etc., which will complement the UK Biobank data collection and contribute to the dementia research in general.
--

VERSION 1 – AUTHOR RESPONSE

Reviewer: 1

Dr. Hugh G. Pemberton, University College London, Deloitte MCS Ltd

Comments to the Author:

This is a well written, timely and relevant study protocol paper. The authors have highlighted the limitations in current assessments for patients with memory concerns and clearly documented their plan to address them. Of particular relevance is the drive to provide MRIs rather than CT scans, where possible. The manuscript could be published in its current state but I have a few comments where further detail or clarity may be beneficial to readers, assuming the authors are willing to make these few changes.

We thank the reviewer for their helpful comments, each of which is address as detailed below.

Introduction

References 1 and 2 could now also include the recent NEJM lecanemab paper (<https://www.nejm.org/doi/pdf/10.1056/NEJMoa2212948>).

We have added the NEJM lecanamab paper

Methods and Analysis

Carer groups are mentioned, are there specific plans to run a support group(s) as part of the Oxford BHC?

We do not currently run support groups, but we do signpost the accompanying relative to local organisations that support carers.

“Patients who require imaging but cannot be referred to the BHC are offered a CT scan as standard”

– What sort of patients cannot be referred to the BHC? In this context, is this just those contraindicated, too frail, or unable to travel for MRI? Please clarify.

Our objective was to offer the BHC visit to as many patients as possible. We have edited the participant selection section to include more detail:

“All patients requiring imaging are referred to the BHC for an MRI scan unless the duty psychiatrist has reason to believe they would not be able to undergo an MRI scan. Reasons include clear contraindications to MRI (implanted devices, metallic foreign bodies, eye injuries, or exceeding the size and weight limitations) and/or patients having mobility problems (limited ability to self-transfer onto the scanner or to lie flat) or being too physically frail to tolerate the length of the BHC appointment. Patients who require imaging but cannot be referred to the BHC are offered a CT scan as standard.”

I note the modified version of the UKB image processing pipeline, could the authors provide more detail here, i.e. will the MRIs be automatically post processed by any research (or commercial) software for further analysis? i.e. WMH quantification, volumetry etc. Will these results be added to the research database?

Our accompanying MRI protocol paper which details the acquisition and analysis methodology and gives links to protocol and data sharing sites has now been published and has been added as a reference (Griffanti et al., 2022; <https://pubmed.ncbi.nlm.nih.gov/36451375/>).

Very encouraging MRI tolerance levels and research uptake for the pilot patients! Do the authors have any comments as per the high research uptake levels and what can be done to replicate this in other initiatives?

Thank you for this observation. The format of this protocol paper (i.e. no discussion) makes it difficult to add a section to detail why we think this model works well, but for your interest we think that we that offering patients a choice of consenting options (rather than just yes or no) is important. We also think our consenting process helps – i.e. we include an ‘executive summary’ of our BHC participant information sheet before patients arrive, so they’re already pre-warned, then our RA’s take time to go through the options at the start of the visit. We ask all patients about their experience as they leave the BHC and have very high rates of patient satisfaction. 92% of patients thought the amount of time they were at the BHC was acceptable and 78% would have been happy to do more during their appointment if asked.

The authors state in the “data sharing” section, that all data will be made available pending request approval etc. It could be useful to state this earlier in the paper (perhaps expand on “article summary” bullet point 4? This currently does make clear that the database will be made available etc.) and/or reference this section earlier, as I suspect this will be of great interest to most readers.

We have added ‘data sharing’ to the summary point 4 as suggested. Note also that data sharing is explicitly described in the abstract.

Reviewer: 2

Prof. Elizabeta Mukaetova-Ladinska, University of Leicester

Comments to the Author:

Well and extensively presented protocol for engaging dementia sufferers in research, creation of data base, clinical and neuroradiological investigations, genotyping etc., which will complement the UK Biobank data collection and contribute to the dementia research in general.

We thank the reviewer for their comments.

Reviewer: 1

Competing interests of Reviewer: n/a

Reviewer: 2

Competing interests of Reviewer: None